# Epidemiology of hepatitis B and C virus infections among patients who booked for surgical procedures at Felegehiwot referral hospital, Northwest Ethiopia

Mulusew Andualem Asemahagn⬛*

School of Public Health, College of Medicine and Health Sciences, Bahir Dar University, Bahir Dar, Ethiopia

* muler.hi@gmail.com

## Abstract

### Background

Hepatitis B virus(HBV) and hepatitis C virus(HCV) are the main causes of cirrhosis, liver cancer, and death. This study aimed to determine the seroprevalence and associated factors of HBV surface antigen(HBsAg) and anti-HCV among patients screened for surgery at Felegehiwot referral hospital, Northwest Ethiopia.

### Methods

A hospital-based cross-sectional study was conducted among 433 patients in 2018. Data on socio-demographic and risk factors were collected by an exit interview using a pretested structured questionnaire. A venous blood sample of 5ml was collected from each participant, and serum was tested for HBsAg and anti-HCV using one-step rapid test kits and enzyme-linked immunosorbent assay. Multivariable logistic regression analysis was computed to identify factors associated with HBV and HCV infections. The odds ratio with 95% CI was used to describe the strength of association and statistical significance.

### Results

A total of 422 patients gave data and included in the analysis. The mean age of patients was 36±5 years. About two-thirds, 269(64%) and 274(65%) patients were males, and from rural areas, respectively. The seroprevalence of HBsAg was 34(8%) followed by 18(4.3%) anti-HCV and 4(0.9%) co-infections. Being single(AOR = 1.96, 95%CI = 1.12–3.10), rural residence (AOR = 2.68, 95%CI = 1.28–5.61), ever heard about HBV (AOR = 2.41, 95%CI = 1.18–5.20), having multiple sexual partners(AOR = 2.85, 95%CI = 1.30–5.58), HIV positive (AOR = 3.14, 95%CI = 1.31–7.61), history of tooth extraction(AOR = 3.0, 95%CI = 1.40–6.56), hospitalization history(AOR = 2.95, 95%CI = 1.26–5.81), sharing of sharp instruments (AOR = 3.86, 95%CI = 1.82–8.79), and had blood contact(AOR = 2.64, 95%CI = 1.14–5.42) were statistically significant factors to HBV infection. Similarly, sharing of sharp instruments (AOR = 4.65, 95%CI = 1.32–15.1), tooth extraction practice(AOR = 2.81, 95%CI = 1.12–

**Funding:** Bahir Dar University covered the budget, but it has no technical role in the research processes.

**Competing interests:** The authors declared that they have no financial and non-financial competing interests in this work.

6.56), surgical history (AOR = 3.68, 95%CI = 1.64–9.82), hospitalization history(AOR = 4.51, 95%CI = 1.62–8.35) and had blood contact(AOR = 3.2, 95%CI = 1.56–8.51) were significant factors to HCV infection.

## Conclusion

The seroprevalence of HBsAg and ant-HCV was high compared to WHO and previous study findings. Giving special attention to awareness creation, rural settings, improving personal behaviors, infection prevention activities of health facilities, quality of healthcare procedures is crucial to prevent viral hepatitis infection.

## Introduction

Viral hepatitis, caused by five types of hepatotropic virus (A—E), becomes one of the world's infectious diseases causing acute and chronic hepatitis that leads to cirrhosis, liver cancer, and death[1, 2]. Viral hepatitis is a serious bloodborne, sexually, vertically, and feco-orally transmitted systemic communicable disease[1, 3]. It can also be transmitted through the reuse of inadequately sterilized medical equipment, surgical procedures, and tooth extraction practices [4–6]. HBV and HCV are responsible for 96% of all hepatitis mortalities worldwide[7]. Globally, over 257 and 71 million people were living with chronic HBVand HCV infections in 2015, respectively[1, 2]. The WHO African Region and the Western Pacific Region accounted for 68% of the global HBV epidemic. Globally, about 2.3 million people have both HBV and HCV co-infections[1]. The prevalence of HBV infection varied from high($\geq$ 8%) to intermediate(2–7%) and low(<2%). Likewise, HCV infection is high(>3.5%), moderate (1.5–3.5%) and low(<1.5%) prevalence[8].

The African countries accounted for the second-largest number of chronic hepatitis carriers next to Asian countries[9]. There were over 60 million HBV carriers in sub-Saharan Africa [10] and the prevalence of HBsAg ranges from 10–20%[11]. Similarly, Ethiopia is one of the high hepatitis epidemic countries as per the WHO classification criteria[8, 12]. A study from Addis Ababa supported this fact where the sero-prevalence of HBsAg and anti-HCV were 35.8% and 22.5%, respectively among chronic patients[13]. A recent study from Southern Ethiopia also showed 9% HBV and 5.5% HCV prevalence among surgical patients[6]. Moreover, a systematic meta-analysis in Ethiopia reported a 6% pooled prevalence of HBV[14]. Based on study findings from Ethiopian pregnant women, the prevalence of HBsAg ranges from 2.4%[15] to 6.3%[16]. Similarly, the prevalence of anti-HCV ranges from 1.8%[3] to 8.0% [15, 17]. The increased surgical practices, injuries, blood transfusion, and unsafe healthcare practices might lead to high hepatitis infection in Ethiopia.

The prevalence of HBsAg and anti-HCV is high in hospitalized surgical patients[1, 2, 7, 8, 13]. However, there is limited evidence about the magnitude of viral hepatitis at the community level and during patient screening for surgery in Ethiopia. Therefore, this study aimed to assess the sero-prevalence of HBsAg and anti-HCV and associated factors among patients who booked for surgery at Feleghiwot referral hospital, Bahir Dar city, Northwest Ethiopia. The findings of this study will be important to health planners, hospital managers, and health workers to know the magnitude of HBV and HCV infections and make informed decisions to fight hepatitis infection.

## Materials and methods

### Study design and setting

A hospital-based cross-sectional study was conducted at Felegehiwot referral hospital located in Bahir Dar City, Northwest Ethiopia. The hospital is established in 1963 and has Emergency, Surgery, Medical, Pediatrics, Obstetrics and Gynecology, Psychiatry, Dental, and Orthopedics units with outpatient, inpatient departments, and follow-up departments. It has about 415 beds and 600 staff in different departments. It has provided its specialized services for over seven million people living in its catchment area. It also offers referral services coming from different parts of the Amhara Region and neighboring regions[18].

### Source and study population

All patients visiting the surgical ward in 2018 were the source population. Whereas, all patients who prescreened and booked for surgical procedures in the medical ward within the specified period were study population to this study.

### Sample size determination and sampling procedures

The sample size of this study was determined using Epi Info version 7 using the following assumptions: 35.8% HBV proportion (P)[13], 5% precision error, and 97% confidence level. Then, the final sample size became 433. A systematic random sampling technique (every other) was applied to select each participant.

### Data collection tool and techniques

Two data sources were included in this study: data on socio-demographic and associated risk factors, and data about HBV and HCV prevalence from blood samples. A pretested structured questionnaire was used to collect data on socio-demographic and risk factors (S1 Table). The questionnaire was validated for its consistency by using Cronbach's alpha test and the value was ($\alpha = 0.87$). Two trained health officers collected data through a face-to-face exit interview. Also, the venous blood sample of 5ml was collected from each study participant by a well-trained laboratory technologist. Then, the serum was separated by centrifugation at 5000 r/min for 15 minutes and tested for HBsAg, anti- HCV using one step HBsAg test strip (Nantong Diagnosis Technology Co., Ltd., China) and one step HCV test strip (Nantong Diagnosis Technology Co., Ltd., China), respectively following the instructions of the manufacturer. The sensitivity and specificity of rapid test Kits were 99.1% and 99.6%, respectively[19]. Then, the samples were further tested using enzyme-linked immunosorbent assay(ELISA) (Dialab GmbH, Wiener Neudorf, Austria) machine as per the manufacturer's manual[20]. Sample collection and process were conducted based on the WHO[21] and Ethiopian national laboratory guidelines for specimen collection, processing, and handling[22].

### Data quality assurance

The questionnaire was pretested on a 5% sample, and validated by Cronbach's alpha test before the actual data collection. Data collectors were trained for two days with practice before data collection. Regular supportive supervision was given to data collectors. The WHO and national guidelines were used to collect, process, and test serum samples. Data completeness and consistency were checked daily, during data entry and analysis.

## Data management and analysis

Data were entered, cleaned, and analyzed using SPSS version 25 software. In this analysis, single marital status means living alone and it is the sum of not married, divorced, and widowed. Descriptive statistics including proportions, the measure of central tendency, and cross-tabulations were computed to describe study participants and state the prevalence of HBsAg, anti-HCV. Bivariate and multivariable logistic regression analyses were used to identify risk factors associated with HBV and HCV infections. Variables with a p-value of less than 0.2 at bivariate analysis were considered to fit the final multivariable logistic regression model through a stepwise (forward) method. Odds ratio with 95%CI and p-value less than 0.05 were used to describe the strength of association and level of significance.

## Ethical clearance

The study was conducted as per the Helsinki Declaration for biomedical research. The study was ethically cleared and approved by the ethical review committee of Amhara Regional Health Bureau (Protocol No: Re/TS/RTT/01/367/09) with a supporting letter to Felegehiwot referral hospital. Data were collected after getting approval from the hospital medical director and written informed consent from each study participant. Participation in the interview and blood sample collection was fully voluntary based. Data confidentiality was kept through avoiding personal identifiers and anonymity of personal data records.

# Results

## Socio-demographic and behavioral variables of study participants

Of the total 433 samples, 422 surgical patients provided complete data and included in the analysis with a response rate of 97.5%. About two-third, 269(64%), 274(65%), and 266(63%) of the study participants were male, rural residents and in union marital status, respectively. The mean and standard deviation age was 36 ±5 years. Over a third, 152(36%) of the respondents had no formal education. Over half, 249(59%) of the respondents were farmers and 102(24%) had more than one sexual partner. Only 110(26%) respondents ever heard about HBV infection. About 25(5.9%) and 36(8.5%) respondents were HIV positive and had blood transfusion history, respectively. Sixty-two(15%) patients had tooth extraction practices. Also, 42(10%) patients had hospitalization history and 28(6.6%) had a history of surgical procedures. Moreover, 38(9%) of the patients had blood contact history with unknow status in their life (Table 1).

## Sero-prevalence of HBV and HCV infections

The overall prevalence of HBsAg and anti-HCV were 34(8%) [95% CI = 5.6–10.4%], and 18 (4.3%) [95%CI = 3.0–6.2%], respectively. Four of the infected respondents (0.9%) had HBV and HCV coinfection. The positivity rate for both infections was relatively higher among males, people above 36 years, rural residents, people who shared sharp instruments, who had no history of blood transfusion and surgery procedures (Tables 1 and 2).

## Factors associated with HBV infection

Based on the multivariable logistic regression model, patients from rural areas were over double times more likely to have HBV infection compared to patients from urban areas (AOR = 2.68, 95% CI = 1.28–5.61). Similarly, patients with single marital status and ever heard about HBV were twice to get HBV infection than patients in union and who did not hear about HBV(AOR = 1.96, 95%CI = 1.12–3.10, and AOR = 2.41, 95%CI = 1.18–5.20),

**Table 1. Factors of HBV infection on surgical patients in Felegehiwot referral hospital, 2018.**

| Variables | HBsAg status | | COR (95%CI) | AOR (95%CI) |
|---|---|---|---|---|
| | Positive (%) | Negative (%) | | |
| **Age in years** | | | | |
| ≤ 36 | 9 (2.0) | 110 (26.1) | 0.88 (0.40–1.94) | - - |
| > 36 | 25 (6.0) | 278 (65.9) | 1 | |
| **Sex** | | | | |
| Male | 18 (4.2) | 251 (59.5) | 0.61 (0.30–1.24) | - - |
| Female | 16 (3.8) | 137 (32.5) | 1 | |
| **Residence** | | | | |
| Rural | 28 (6.6) | 246 (58.3) | 2.80 (1.18–6.64) | 2.68 (1.28–5.61) |
| Urban | 6(1.4) | 142 (33.7) | 1 | 1 |
| **Marital status** | | | | |
| Single | 19 (4.5) | 137 (32.5) | 2.32 (1.14–4.71) | 1.96 (1.12–3.10) |
| In union | 15 (3.5) | 251 (59.5) | 1 | 1 |
| **Education level** | | | | |
| No formal education | 10 (2.4) | 142 (33.7) | 0.72 (0.34–1.60) | - - |
| Formal education | 24 (5.6) | 246 (58.3) | 1 | |
| **Occupation** | | | | |
| Farmer | 11 (2.6) | 139 (33.0) | 0.67 (0.30–1.55) | - - |
| Non-employed | 15 (3.5) | 130 (31.0) | 1.71 (0.70–4.20) | - - |
| Employed | 8 (1.9) | 119 (28.0) | 1 | |
| **HIV sera-status** | | | | |
| Positive | 6 (1.4) | 19 (4.5) | 4.16 (1.54–11.3) | 3.14 (1.31–7.61) |
| Negative | 28 (6.6) | 369 (87.5) | 1 | 1 |
| **Ever hear about HBV** | | | | |
| Yes | 16 (3.8) | 94 (22.3) | 2.78 (1.36–5.67) | 2.41 (1.18–5.20) |
| No | 18 (4.3) | 294 (69.7) | 1 | 1 |
| **History of multiple sexual partners** | | | | |
| Yes | 16 (3.8) | 86 (20.4)302 | 3.12 (1.53–6.40) | 2.85 (1.30–5.58) |
| No | 18 (4.2) | (71.6) | 1 | 1 |
| **Sharing of sharp instruments** | | | | |
| Yes | 7(1.6) | 18 (4.3) | 5.33 (2.05–13.87) | 3.86 (1.82–8.79) |
| No | 27 (6.4) | 370 (87.7) | 1 | 1 |
| **History of blood transfusion** | | | | |
| Yes | 5 (1.2) | 31 (7.3) | 1.99 (0.72–5.50) | - - |
| No | 29 (6.9) | 357 (84.6) | 1 | |
| **Tooth extraction practice** | | | | |
| Yes | 10 (2.4) | 52 (12.3) | 2.70 (1.22–5.95) | 3.0 (1.40–6.56) |
| No | 24 (5.7) | 336 (79.6) | 1 | 1 |
| **History of hospitalization** | | | | |
| Yes | 8 (1.9) | 34 (8.0) | 3.20 (1.35–7.62) | 2.95 (1.26–5.81) |
| No | 26 (6.2) | 354 (83.9) | 1 | 1 |
| **History of surgical procedures** | | | | |
| Yes | 5 (1.2) | 23 (5.4) | 2.73 (0.89–7.73) | - - |
| No | 29 (6.9) | 365(86.5) | 1 | |
| **History of direct blood contact** | | | | |
| Yes | 7 (1.6) | 31 (7.3) | 2.98 (1.20–7.41) | 2.64 (1.14–5.42) |
| No | 27 (6.4) | 357 (84.6) | 1 | 1 |

(*Continued*)

**Table 1.** (Continued)

| Variables | HBsAg status | | COR (95%CI) | AOR (95%CI) |
|---|---|---|---|---|
| | **Positive (%)** | **Negative (%)** | | |
| **History of needle stick anywhere** | | | | |
| Yes | 6 (1.4) | 58 (13.8) | 1.22 (0.48–3.10) | - - |
| No | 28 (6.6) | 330 (78.2) | 1 | |

respectively. Also, patients who practiced sharing of sharp instruments, tooth extraction, and had HIV positive status were over three times more likely to acquire HBV infection compared to their counterparts (AOR = 3.86, 95%CI = 1.82–8.79; AOR = 3.0, 95%CI = 1.40–6.56; AOR = 3.14, 95%CI = 1.31–7.61), respectively. Moreover, the odds of having HBV infection were 2.85 times more among patients who had a history of multiple sexual practices compared to patients who had no history (AOR = 2.85, 95%CI = 1.30–5.58). Patients who had hospitalization history were 2.95 times more likely to get HBV infection than patients with no hospitalized history (AOR = 2.95, 95%CI = 1.26–5.81). Also, patients who encountered direct blood contact were over two-folds to have HBV infection compared to patients with no blood contact history(AOR = 2.64, 95%CI = 1.14–5.42) (Table 1).

## Factors associated with HCV infection

In the multivariable logistic regression analysis, the odds of getting HCV infection were four times more among patients who shared sharp instruments and had hospitalization history compared to the counterpart patients(AOR = 4.65, 95%CI = 1.32–15.1, and AOR = 4.51, 95% CI = 1.62–8.35), respectively. Similarly, patients with surgery history were over three times to have HCV infection than patients who had no history of surgery (AOR = 3.68, 95%CI = 1.64–9.82). Also, patients with blood contact and tooth extraction history were more likely to have HCV infection than patients with no history of blood contact and tooth extraction (Table 2).

## Discussion

This study tried to assess the magnitude of HBV and HCV infections among patients who are going to have surgery at Felegehiwot Referral hospital. The current prevalence of HBV(8%) and HCV(4.3%) (Table 1) are high according to WHO classification criteria[8]. This finding was slightly in agreement with study findings from Ethiopian studies in Addis Ababa[23], Hawasa[6], and Gambella[24], where HBV prevalence was 7.8%, 9%, and 7.9%, respectively.

The current HBV and HCV prevalence were, however, found to be higher compared to study findings from Bahir Dar city among blood donors that reported 3.9% HBV and 0.6% HCV prevalence[25]. These discrepancies might be attributed to differences in age, sample size, study participants (patient vs healthy blood donors), and study design. Residence might also contribute to this variation where most of the patients were from different parts of the Amhara region, while the blood donors are from Bahir Dar city and adjacent districts.

Similarly, this finding was also slightly higher compared to study findings among community members from Southern Ethiopia[26] that revealed 7.2% HBV and 1.9% HCV sero-prevalence. This variation might be related to differences in study participants (chronic patients vs community members), and diagnostic methods used to report participants' status; for example, the current study used ELISA as a confirmatory test which was not applied by the previous study.

**Table 2. Factors of HCV on surgical patients in Felegehiwot referral hospital, 2018.**

| Variables | Anti-HCV status | | COR (95%CI) | AOR (95%CI) |
|---|---|---|---|---|
| | Positive (%) | Negative (%) | | |
| **Age in years** | | | | |
| ≤ 36 | 6 (1.4) | 113 (26.8) | 1.30 (0.47–3.51) | - - |
| > 36 | 12 (2.8) | 291 (70.0) | 1 | |
| **Sex** | | | | |
| Male | 10(2.4) | 261 (61.8) | 0.66 (0.26–1.80) | - - |
| Female | 8 (1.9) | 143 (33.9) | 1 | |
| **Residence** | | | | |
| Rural | 8 (1.9) | 266 (63) | 0.42 (0.16–1.07) | - - |
| Urban | 10(2.4) | 138 (32.7) | 1 | |
| **Marital status** | | | | |
| Single | 6 (1.4) | 150 (35.6) | 0.85 (0.31–2.30) | - - |
| In union | 12 (2.8) | 254 (60.2) | 1 | |
| **Education level** | | | | |
| No formal education | 7 (1.6) | 145 (34.3) | 1.14 (0.43–3.00) | - - |
| Formal education | 11 (2.6) | 259 (61.4) | 1 | |
| **Occupation** | | | | |
| Farmer | 5 (1.2) | 145 (34.3) | 0.70 (0.21–2.33) | - - |
| Non-employed | 7 (1.7) | 138 (32.7) | 1.02 (0.33–3.13) | - - |
| Employed | 6 (1.4) | 121 (28.7) | 1 | |
| **History of multiple sexual partners** | | | | - - |
| Yes | 7 (1.7) | 95 (22.5) | 2.10 (0.78–5.50) | |
| No | 11 (2.6) | 309 (73.2) | 1 | |
| **Sharing of sharp instruments** | | | | |
| Yes | 6 (1.4) | 19 (4.5) | 9.50 (3.43–29.5) | 4.65 (1.32–15.1) |
| No | 12 (2.8) | 385 (91.3) | 1 | 1 |
| **History of blood transfusion** | | | | |
| Yes | 3 (0.7) | 36 (8.5) | 2.00 (0.57–7.40) | - - |
| No | 15 (3.6) | 368 (87.2) | 1 | |
| **Tooth extraction practice** | | | | |
| Yes | 6 (1.4) | 56 (13.3) | 3.11 (1.12–8.62) | 2.81 (1.11–6.56) |
| No | 12 (2.8) | 348 (82.5) | 1 | 1 |
| **History of hospitalization** | | | | |
| Yes | 7 (1.7) | 35 (8.3) | 6.71 (2.45–18.4) | 4.51 (1.62–8.35) |
| No | 11 (2.6) | 369 (87.4) | 1 | 1 |
| **History of surgical procedures** | | | | |
| Yes | 5 (1.2) | 23(5.4) | 6.4 (2.10–19.0) | 3.68 (1.64–9.82) |
| No | 13 (3.1) | 381(90.3) | 1 | 1 |
| **History of direct blood contact** | | | | |
| Yes | 6 (1.4) | 32 (7.6) | 5.81 (2.10–16.51) | 3.2 (1.56–8.51) |
| No | 12 (2.8) | 372 (88.2) | 1 | 1 |
| **History of needle stick** | | | | |
| Yes | 5 (1.2) | 59 (14.0) | 2.25 (0.77–6.54) | - - |
| No | 13 (3.1) | 345 (81.7) | 1 | |

Besides, this HBV prevalence was found to be higher compared to study findings among pregnant women attending antenatal care services in Ethiopia; 2.4% from Oromia region[15], 4.5% from Southern Ethiopia[3] and 6.3% from Harar[16]. These discrepancies might be related to differences in residence, health status (pregnant women vs chronic patients), and the research methods used (using only rapid kits vs both rapid kits and ELISA). It is also higher than findings from a systematic study in Ethiopia that revealed a 6% pooled prevalence[14]. This difference might be attributed to variations in facility type (health centers vs referral hospital), gender where most papers reported higher prevalence among males[7, 13, 27], and personal behaviors between females and males.

This finding on the other hand was lower compared with study findings among chronic patients in Addis Ababa Ethiopia[13] where HBV and HCV prevalence were 35.8% and 22.5%, respectively. This discordant might be related to differences in sample size(129 vs 422), study period(2011 vs 2018), and study area where the former study was among patients from Black lion, Set Pawul, and Zewuditu memorial hospitals which are national specialized hospital serving as referral sites to Ethiopia. Thus, viral hepatitis will be higher in those hospitals compared to Felegehiwot referral hospital since more chronic cases will be referred to those national hospitals.

Also, the present HBV prevalence was found to be lower than a pooled prevalence from a systematic meta-analysis study in Sudan in which HBV was 12%[27]. This might be due to research design where the systematic review included more research works having various prevalence. But in our study, it is a single study from one hospital, thus the prevalence is lower.

Concerning HCV infection, the present HCV prevalence was higher than a pooled prevalence of HCV(2.74%)[27]. This discrepancy might be attributed to variations in study design (survey vs meta-analysis), study settings, and population behaviors. Similarly, it was higher than study findings from Southern Ethiopia(1.8%)[3] and Bahir Dar city(3.8%)[17]. These variations might be due to differences in the history of hospitalization, blood transfusion, unsafe healthcare procedures, and traditional practices.

On the contrary, the prevalence of HCV was found to be lower compared to a study finding from Hawasa where HCV prevalence was 5.5%[6]. It was also lower compared to study findings from the Oromia region[15, 28] and Hawasa referral hospital[6] where HCV prevalence were 8% and 9%, respectively.

Based on this study, the positivity rate of HBV and HCV was higher among males and elders ($\geq$36years) although not statistically significant in the multivariable logistic regression model. The previous studies also supported this finding[13, 17, 29]. This might be due to variations in genetic make-up[30], status of immune system, exposure to sexual practices, history of health facility visit, tooth extraction practices, and history of blood transfusion.

In high hepatitis epidemic areas, the co-infection of HBV and HCV is reported since the two viruses share common routes of transmission[6, 26, 31, 32]. Similarly, 0.9% co-infection of HBV and HCV was reported in this study. This finding was lower compared to study findings from Ethiopia[13], India[31], and Mongolia[32] where 2.5%, 5.9%, and 7.7% co-infections of HBV and HCV were reported among chronic patients, respectively. Differently, it was relatively higher than study finings from Southern Ethiopia[6, 26] which reported 0.2% and 0.3% co-infections. This might be due to differences in behaviors of study participants, geographic locations, and types of diagnostic tools used to screen out the study participants. For example, the two studies from Southern Ethiopia did not use ELISA as a confirmatory test which was applied in our study.

This study identified various risk factors for HBV and HCV infections (Tables 1 and 2). Patients from rural residences were over twice to have HBV infection. This implies the presence of high hepatitis infections among rural communities with unknown status as a result of

poor screening practices. This might be related to low community awareness, limited access to healthcare facilities, less precaution to blood and body fluids, sharing of sharp instruments, and high prevalence of traditional practices including home-based delivery, tooth extraction, and female genital mutilation in rural areas[3, 28]. Similarly, being single in marital status was found to be an important factor to acquire HBV infection. In this analysis, single marital status is to mean the sum of number of patients who are divorced, widowed and not married. Thus, these people might have sexual exposure to more than one partner whose hepatitis status is unknown compared to patients who are in union.

Having a history of multiple sexual partners was an important risk factor to acquire HBV infection. This indicates the presence of unsafe sexual practices within the community which is a serious practice to expose people not only to hepatitis but also to HIV and other sexually transmitted infections. Similar study findings from Ethiopia and abroad also supported this finding[3, 6, 16, 24, 27, 28, 33, 34]. This is because of unsafe sexual practices with more partners whose status is unknown since blood and body fluids are key infection sources. Likewise, the odds of being positive to HBsAg was triple times among HIV positive patients. This will make the situation more serious among surgical patients due to more comorbidities (HBV, HIV, and surgery cases). This was also reported by the previous studies from Ethiopia[16, 25]. This might be due to having multiple sexual partners with unsafe sexual practices. Both hepatitis and HIV are bloodborne diseases transmitted by sexual contact, direct blood contact, through blood-contaminated equipment, and mother-to-child during pregnancy and delivery [1, 3, 7].

Moreover, history of hospitalization, tooth extraction practice, sharing sharp instruments, and direct contact to blood with unknown status were statistically significant factors to HBsAg and anti-HCV(Tables 1 and 2). This implies that hospital services and tooth management practices are unsafe and main sources of infections for patients coming for healthcare services. It is to mean patients have a chance to get infections other than the disease that makes them visit health facilities[1, 2] which is a serious issue that requires special attention from the health bureau and health facilities. This was in agreement with previous study findings that reported these variables as risk factors to hepatitis infection[6, 13, 16, 35]. This could be due to poor infection control practices in health facilities, using poorly sterilize medical equipment, improper healthcare procedures, traditional tooth extraction practices, sharing sharp instruments, blood contact with bare hand[1–4]. Tooth extraction, reuse of equipment without proper sterilization, and sharing sharp instruments (blade, needle, pin, syringe. . .) are common among rural communities in Ethiopia[13, 16]. All these can facilitate the transmission of HBV and HCV infections[1–4, 35] and might be happened due to poor infection prevention practices, absence of protective equipment, and low awareness of the community and traditional practitioners.

Patients with surgical history were triple times more likely to have HCV infection. This implies that health facilities are key sources of hepatitis infection. Therefore, evaluating the quality of healthcare services and procedures, application of healthcare ethics, presence and adherence of infection prevention protocols, and accountability and awareness of health workers is needed. Previous studies also reported the same finding on patients' history of surgical procedures and HCV infection[2, 6, 23, 35]. In line with this, patients who shared sharp instruments with others were 4.65 times more likely to get HCV infection compared to the counterpart patients.

Based on the hepatitis management guideline and literature, a history of blood transfusion was statistically significant factor to HBV and HCV infections[15, 25, 28]. Blood transfusion with inadequate screening can indeed transmit blood-borne diseases. Also, due to the absence of quality screening tools at the window periods, blood screening results might be negative.

Thus, blood transfusions based on such screening results can transmit infections. Unlike the previous studies[1–3, 16, 28], history of blood transfusion was not statistically significant to HBV and HCV infections in this study. This might be due to a small number of patients who had blood transfusion history due to low awareness, attention, and practice of blood transfusion.

## Conclusions

This study depicted a high prevalence of HBV and HCV infection among patients who screened for surgical procedures. The positivity rate was higher among male, rural residents, and elder patients. Being from rural areas, single marital status, having HIV infection, sharing sharp instruments, history of hospitalization, tooth extraction practices, and blood contact with bare hand were risk factors of HBV infection. Similarly, history of hospitalization, surgical procedures, blood contact, and sharing sharp instruments were risk factors of HCV infection. Improving awareness of community and health workers, access to HBV and HCV screening services, enhancing infection prevention practices, adherence to healthcare service guidelines, and practicing medical ethics are important to prevent acquiring HBV and HCV infections.

## Supporting information

**S1 Table. Personal descriptions and risk assessment questionnaire.**
(PDF)

**S1 Dataset. HBV and HCV data set.**
(SAV)

## Acknowledgments

The author would like to express his deepest gratitude to the Amhara regional health bureau for giving ethical clearance and supporting letter, and to the staff of Felegehiwot referral hospital, data collectors, and study participants for their candid supports during data collection.

## Author Contributions

**Conceptualization:** Mulusew Andualem Asemahagn.

**Data curation:** Mulusew Andualem Asemahagn.

**Formal analysis:** Mulusew Andualem Asemahagn.

**Funding acquisition:** Mulusew Andualem Asemahagn.

**Investigation:** Mulusew Andualem Asemahagn.

**Methodology:** Mulusew Andualem Asemahagn.

**Project administration:** Mulusew Andualem Asemahagn.

**Resources:** Mulusew Andualem Asemahagn.

**Software:** Mulusew Andualem Asemahagn.

**Supervision:** Mulusew Andualem Asemahagn.

**Validation:** Mulusew Andualem Asemahagn.

**Visualization:** Mulusew Andualem Asemahagn.

**Writing – original draft:** Mulusew Andualem Asemahagn.

**Writing – review & editing:** Mulusew Andualem Asemahagn.

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
