## [Decision Letter · Decision Letter 0]

2 Apr 2020

PONE-D-20-06821

Epidemiology of hepatitis B and C virus infections among patients who booked for surgical procedures at Felegehiwot referral hospital, Northwest Ethiopia

PLOS ONE

Dear Dr Asemahagn,

Thank you for submitting your manuscript to PLOS ONE. After careful consideration, we feel that it has merit but does not fully meet PLOS ONE’s publication criteria as it currently stands. Therefore, we invite you to submit a revised version of the manuscript that addresses the minor points raised during the review process.

We would appreciate receiving your revised manuscript by 4 weeks. To enhance the reproducibility of your results, we recommend that if applicable you deposit your laboratory protocols in protocols.io, where a protocol can be assigned its own identifier (DOI) such that it can be cited independently in the future. For instructions see: http://journals.plos.org/plosone/s/submission-guidelines#loc-laboratory-protocols

We look forward to receiving your revised manuscript.

Kind regards,

Isabelle Chemin, PhD

Academic Editor

PLOS ONE

Journal Requirements:

2. Please include additional information regarding the survey or questionnaire used in the study and ensure that you have provided sufficient details that others could replicate the analyses. For instance, if you developed a questionnaire as part of this study and it is not under a copyright more restrictive than CC-BY, please include a copy, in both the original language and English, as Supporting Information. Moreover, please include more details on how the questionnaire was pre-tested, and whether it was validated.

- Taye, Meseret, et al. "Magnitude of hepatitis B and C virus infections and associated factors among patients scheduled for surgery at Hawassa University comprehensive specialized Hospital, Hawassa City, southern Ethiopia." BMC research notes 12.1 (2019): 412.

The text that needs to be addressed involves some sentences of the Introduction

In your revision ensure you cite all your sources , and quote or rephrase any duplicated text outside the methods section. Further consideration is dependent on these concerns being addressed.

4. As we note that data collection was performed by a cooperative group, we would recommend that you consult our authorship requirements page: https://journals.plos.org/plosone/s/authorship, to ensure that everyone who meets our criteria for authorship  is listed as an author.

Reviewers' comments:

Reviewer's Responses to Questions

**Comments to the Author**

1. Is the manuscript technically sound, and do the data support the conclusions?

Reviewer #1: Yes

2. Has the statistical analysis been performed appropriately and rigorously? 

Reviewer #1: Yes

3. Have the authors made all data underlying the findings in their manuscript fully available?

Reviewer #1: Yes

4. Is the manuscript presented in an intelligible fashion and written in standard English?

Reviewer #1: Yes

5. Review Comments to the Author

Reviewer #1: The paper by Pr Asemahagn (PONE-D-20-06821) describes the prevalence of HBsAg and anti-HCV in patients enrolled for surgery in a referral hospital in Northwest Ethiopia.

Major remark:

The Author points out that a study population of blood donors means that most participants come from the large city of the region. In this case the study population comes from all parts of the Amhara region and even some from neighboring regions. However, one can wonder whether the population studied in this paper is really representative of the general population. There are 2 major biases. As could be expected for a group enrolled for surgery, the patients tend to be older, with more than 70% being older than 36 years. There is also a gender bias, with almost two-thirds of the patients being male. These limitations could be discussed in more detail.

Minor remarks:

1) The names of the kits and the manufacturers of the kits used for detecting HBsAg and anti-HCV should be given.

2) Page 5, line 1. “suppression” is certainly not what the Author meant to say. Could it be “supervision”?

3) Also page 5 in the Ethical clearance section, the Author writes “getting ok”. This is too colloquial. “getting permission” would be better.

4) Throughout the paper the Author talks about “sharps”. This is imprecise. “sharp tools” or “sharp instruments” should be used.

5) In Table 1, for tooth extraction/HBsAg negative, the % are missing

6) In Table 2, the value 121 on the line non-employed/anti-HCV negative belongs on the line below.

7) Page 9, line 5. It should be “high” and not “higher”

8) Page 9, second from last line. It should be “from”

9) Page 11, first line. It should be “odds”

10) Page 11, line 3. It should be “previous” instead of “former”.

11) Page 13. Since there is only 1 Author, the Competing interests section should read “The Author declares that he has no competing interests in this work”. Similarly, in the Author’s contributions section it should read “MAA did the conception …………………….interpretation of data and drafting of the manuscript”

12) In references 5 and 6 “et al” should be spelled out in full and not abbreviated to “ea”

6. PLOS authors have the option to publish the peer review history of their article (what does this mean?). If published, this will include your full peer review and any attached files.

Reviewer #1: Yes: Alan Kay

---

## [Author Response · Author response to Decision Letter 0]

28 Apr 2020

Cover letter to PLOS ONE journal editor

Isabelle Chemin, PhD 216th April 2020

 Academic editor 

 PloS ONE journal

Subject: Submitting review response to Epidemiology of hepatitis B and C virus infections among patients who booked for surgical procedures at Felegehiwot referral hospital, Northwest Ethiopia [PONE-D-20-06821]

 Dear Dr. Isabelle Chemin,

Thank you for your email enclosing the reviewers’ comments and continuous email reminders. I, the authors have great thanks to the journal editor and the reviewers for reviewing and giving insightful comments to our manuscript. I have carefully reviewed the comments, suggestions, clarity questions, and revised the manuscript accordingly. I made significant revisions to all sections of a manuscript for language, clarity, concept, shortening sentences, changing words, adding explanations, incorporating missed findings, avoiding misleading words or phrases, and using appropriate punctuation. For details, please refer the general reflection to the journal editor, and a point by point responses to each reviewer below. In the one by one response, comments of the reviewers are indicated in italics, and replies to the comments are shown by the normal font style. Similarly, changes to the manuscript are shown by track changes in the manuscript. As evidence, we attached the track changed manuscript besides this point by point response.

Note: Activities and strategies that we have used to make this remarkable revision are presented below as a general direction or an introduction to a point by point response.

I hope, the revised version is now suitable for publication and look forward to hearing from you in due course. 

 Sincerely, 

 Mulusew Andualem Asemahagn

 Associate professor of public health

 School of Public Health, CMHS, Bahir Dar University, Ethiopia

Response to Reviewers

General reflections to the PLoS one journal editors

Dear Plos one journal editors and reviewers, I hearty acknowledged your efforts made in each section of our manuscript. I am very happy with all the given comments and clarity questions that you raised. I know that the purpose of the review is improving the quality and readability of the manuscript. I fully accept them and did our possibilities by giving time to each section of a manuscript to address the given comment and clarity questions. My approaches were not focusing on answering only the raised issues, but revising each section by taking the raised issues as triggering points. Frankly speaking, I see this as an excellent opportunity to revise the manuscript and increase its quality. 

Activities that I made are incorporating comments/suggestions, giving answers/explanations/ justifications/reasons/to clarity questions, adding concepts, inserting new words/phrases/ sentences/ paragraphs, deleting less important words/phrases/sentences/paragraphs, improving language problems (type errors, wording, clarity, and sentence parallelism and complexity). Removing unnecessary spaces between words, sentences, paragraphs…are made. NB: The reordering of paragraphs was also made to the discussion section to better idea coherence. I also included new paragraphs to the discussion section that were missed to discuss in the former manuscript (about co-infection). About information to HBV was also included in the abstract and the result sections (Table). The revisions are made to each section. I am sure that most of the clarity questions and comments given by the respected reviewers are addressed as much as possible. 

A. Responses to the academic editor

Comment 1: “1. Please ensure that your manuscript meets PLOS ONE's style requirements, including those for file naming.”

Replay 1: I fully accept your comments and tried to be inline as per the PLoS ONE journal format. 

Comment 2: “Please include additional information regarding the survey or questionnaire used in the study and ensure that you have provided sufficient details that others could replicate the analyses. For instance, if you developed a questionnaire as part of this study and it is not under copyright more restrictive than CC-BY, please include a copy, in both the original language and English, as Supporting Information. Moreover, please include more details on how the questionnaire was pre-tested, and whether it was validated.”

Replay 2: The author fully accepts this valuable comment to adhere to the journal format since it is a key criterion. Based on the comment, I added the questionnaire in both the local and English language format. The questionnaire was developed by referring related papers and based on the author’s study objective. It was validated among patients from other hospital and modifications to clarity and order was made. It was also validated by the Cronbach alpha with a value of 0.87 at SPSS version 25. It is stated on page 4, data collection tool, and technique subheading. 

Comment 3: “We noticed you have some minor occurrence of overlapping text with the following previous publication(s), which needs to be addressed: Taye, Meseret, et al. "Magnitude of hepatitis B and C virus infections and associated factors among patients scheduled for surgery at Hawassa University comprehensive specialized Hospital, Hawassa City, southern Ethiopia." BMC research notes 12.1 (2019): 412. The text that needs to be addressed involves some sentences of the Introduction. In your revision ensure you cite all your sources and quote or rephrase any duplicated text outside the methods section. A further consideration is dependent on these concerns being addressed.”

Replay 3: I thank you much for the concern you raised. Unfortunately, I did not have access to that document during the manuscript preparation. However, there was one line which was matching with that document. Hence, thank you for your notice. The similarity has happened since the two authors used the same reference document to talk about HBC and HCV in Africa where there is limited literature since hepatitis is getting attention late by the WHO and regional governments. Now, I downloaded and cited as literature to my manuscript. Thus, thank you again for showing updated literature. Based on your comment, the introduction section is highly revised using recent WHO and research articles to make it strong. I also included researched for updated literature and got about eight literature important to update my manuscript and increase my references from 24 to 32, which is because of your comment. Thus, paragraph 2 of the introduction, which you commented as it has some similarity without citation from the former study is revised as follows: 

The African countries accounted for the second-largest number of chronic hepatitis carriers next to Asia 9]. There were over 60 million HBV carriers in sub-Saharan Africa [10] and the prevalence of HBsAg ranges from 10-20% [11]. Similarly, Ethiopia is one of the high hepatitis epidemic countries as per the WHO classification criteria [8, 12]. A study from Addis Ababa also supported this fact where HBsAg and anti-HCV were 35.8% and 22.5%, respectively among chronic patients [13]. A recent study from Southern Ethiopia also showed 9% HBV and 5.5% HCV prevalence among surgical patients [6]. A systematic meta-analysis in Ethiopia also reported a 6% pooled prevalence of HBV [14]. Based on study findings from Ethiopian pregnant women, the prevalence of HBsAg ranges from 2.4% [15] to 6.3% [16]. Similarly, the prevalence of anti-HCV ranges from 1.8% [3] to 8.0% [15, 17]. The increased surgical practices, injuries, blood transfusion, and unsafe health practices might lead to hepatitis infection in Ethiopia.

Comment 4: “As we note that data collection was performed by a cooperative group, we would recommend that you consult our authorship requirements page: https://journals.plos.org/plosone/s/authorship, to ensure that everyone who meets our criteria for authorship is listed as an author.”

Replay 4: Thank you for your invitation. All the data collectors were recruited to collect and perform laboratory data with pay. Hence, they did not participate in a fee and did not contribute more other than data collection and processing. 

Comment 5: “In your Data Availability statement, you have not specified where the minimal data set underlying the results described in your manuscript can be found...Upon re-submitting your revised manuscript, please upload your study’s minimal underlying data set as either Supporting Information files or to a stable, public repository and include the relevant URLs, DOIs, or accession numbers within your revised cover letter.”

Replay 5: I thank you very much for your reminder. I have included all the data set and questionnaire. I did not take consent from each study participant to share the data sets to the third party, but, your office urge me to share it. This is my first time to share the private data sets of participants without taking consent to share with the third party. My fear is the reuse of such data set by other researchers without taking consent from the original author. Because, we can not control them since it will be online. Even, students are doing such types of unethical actions to their thesis work without collecting data. It is to mean duplication of others' work with out any efforts. Have you think such issues while asking for data set attachment from authors who got it through various difficulties and the ethics issue. I have encountered with this issue to my student who took the data sets from some where for his research. He did not do any thing except downloading such data set and changing the working places. I have disqualified his attempt, but how about others????? anyways, i fill very much discomfort for two things; sharing my data set without taking consent from each study participants 2). contributing more for false research activities among some authors who are using others' data set from online sources with out any efforts. 

B. Response to Reviewer #1

Dear respected reviewers, I am happy and lucky for your insightful comments and clarity questions in each part of the manuscript. I have accepted all the comments and incorporated them into the manuscript as indicated below and on the track change manuscript. I also did a complete revision to all sections of the manuscript taking your comments as triggering points. 

Comment 1: “The author points out that a study population of blood donors means that most participants come from the large city of the region. In this case, the study population comes from all parts of the Amhara region and even some from neighboring regions. However, one can wonder whether the population studied in this paper is representative of the general population. There are 2 major biases. As could be expected for a group enrolled for surgery, the patients tend to be older, with more than 70% being older than 36 years. There is also a gender bias, with almost two-thirds of the patients being male. These limitations could be discussed in more detail.”

Replay 1: Thank you for raising this important issue. I fully accept and revised the issue that you mentioned regarding representativeness to the community. The author stated as the sample could be relatively representative of the population compared to blood donors by assuming that they are coming from different directions. When the author said this, it is not to mean they are representatives of the community. But, when we compared these findings with findings from the blood donor in Bahir Dar city, it is different. So, as per the author’s view, the variation might be related to differences in residence, health conditions, age groups, sample size, study participants (patient vs blood donors), and study design (survey vs retrospective). Thus, it is not to mean the sample or studied people are representatives of the community. Because they are patients who can not represent healthy people. The majority were males that might not represent the proportioning of the community. Most of the patients were also elders, so they might not be real representatives of the community. Thus, by considering these, I have revised the manuscript not to be understood as you mentioned above. Thus, the author deleted sentences that lead to representatives from the last paragraph of the background on page 3. It is also revised on page 9, discussion section, paragraph 1 and 2. 

Comment 2: “The names of the kits and the manufacturers of the kits used for detecting HBsAg and anti-HCV should be given.” 

Replay 2: I fully accept your comment and mentioned all the names and manufacturers of the rapid kits and ELISA on page4, under data collection tools and techniques section as:

“Then, the serum was separated by centrifugation at 5000 r/min for 15 min and tested for HBsAg, anti- HCV using one step HBsAg test strip (Nantong Diagnose Technology Co., Ltd., China) and one step HCV test strip (Nantong Diagnose Technology Co., Ltd., China), respectively following the instructions of the manufacturer. The sensitivity and specificity of rapid test Kits were 99.1% and 99.6%, respectively[18]. Then, the samples were further tested using enzyme-linked immunosorbent assay (ELISA) (Dialab GmbH, Wiener Neudorf, Austria) machine as per the manufacturer’s manual[19]. Sample collection and process were conducted based on the WHO[20] and Ethiopian national laboratory guidelines for specimen collection, processing, and handling[21].” 

Comment 3: “Page 5, line 1. “suppression” is certainly not what the author meant to say. Could it be “supervision”?

Replay 3: Yes, you are. It is to mean supervision and corrected there. 

Comment 4: “Also page 5 in the Ethical clearance section, the Author writes “getting ok”. This is too colloquial. “getting permission” would be better.”

Replay 4: I fully accept your comment and revised as getting approval from… 

Comment 5: “Throughout the paper, the author talks about “sharps”. This is imprecise. “sharp tools” or “sharp instruments” should be used.”

Replay 5: Thank you for giving this comment and I revised the phrase as “sharp instruments” throughout the document and tables.

Comment 6: “In Table 1, for tooth extraction/HBsAg negative, the % is missing”

Replay 6: Thank you much for showing this editorial comment and corrected as per your comment.

NB. To Table 1, I also include a variable “ever heard about HBV” to assess their rough awareness. 

Comment 7: “In Table 2, the value 121 on the line non-employed/anti-HCV negative belongs on the line below.”

Replay 7: I accept it and corrected it as required. 

Comment 8: “Page 9, line 5. It should be “high” and not “higher”

Replay 8: I accept your comment and I revised it as “high” since the classification criteria say high, intermediate, low…

Comment 9: “Page 9, second from the last line. It should be “from” 

Replay 9: Thank you for your comment. It is revised from “form” to “from”

Comment 10: “Page 11, first line. It should be “odds”

Replay 10: I accept it and corrected it as “odds” throughout the document. 

Comment 11: “Page 11, line 3. It should be “previous” instead of “former”.

Replay 11: It is corrected as “…reported by previous studies from Ethiopia.”

 Note: To the discussion section, page 10, I included a paragraph that discussed HBV and HCV co-infection which was missed in the former manuscript. As a result, new references are also included since we used them to discuss this paragraph. 

Comment 12: “Page 13. Since there is only 1 Author, the Competing interests section should read “The author declares that he has no competing interests in this work”. Similarly, in the Author’s contributions section, it should read “MAA did the conception ……………………interpretation of data and drafting of the manuscript”

Replay 12: I fully accept and correct the indicated points as per the given comments. The conflicts of interest section is revised as “The Author declares that he has no competing interests in this work.” 

Similarly, the author’s contribution section is revised as “MAA did the conception, design of the research, data collection, analysis and interpretation of data and drafting of the manuscript.” 

Comment 13: “In references 5 and 6 “et al” should be spelled out in full and not abbreviated to “ea”

Replay 13: Thank you much. I accept your comment and revised all references that have et.al.

Thank you much for your contributions!

---

## [Decision Letter · Decision Letter 1]

3 Jun 2020

Epidemiology of hepatitis B and C virus infections among patients who booked for surgical procedures at Felegehiwot referral hospital, Northwest Ethiopia

PONE-D-20-06821R1

Dear Dr. Asemahagn,

We’re pleased to inform you that your manuscript has been judged scientifically suitable for publication and will be formally accepted for publication once it meets all outstanding technical requirements.

Kind regards,

Isabelle Chemin, PhD

Academic Editor

PLOS ONE

Additional Editor Comments (optional):

Reviewers' comments:

Reviewer's Responses to Questions

**Comments to the Author**

1. If the authors have adequately addressed your comments raised in a previous round of review and you feel that this manuscript is now acceptable for publication, you may indicate that here to bypass the “Comments to the Author” section, enter your conflict of interest statement in the “Confidential to Editor” section, and submit your "Accept" recommendation.

Reviewer #1: (No Response)

2. Is the manuscript technically sound, and do the data support the conclusions?

Reviewer #1: Yes

3. Has the statistical analysis been performed appropriately and rigorously? 

Reviewer #1: Yes

4. Have the authors made all data underlying the findings in their manuscript fully available?

Reviewer #1: Yes

5. Is the manuscript presented in an intelligible fashion and written in standard English?

Reviewer #1: Yes

6. Review Comments to the Author

Reviewer #1: 1) On lines 207 and 252 it should be "non-agreement" and not "in-agreement"

2) On lines 288 and 312 it should be "bare" and not "bear"

3) In Table 2, in the History of multiple sexual partners section, all the numbers should be brought down one line to match up with the Yes and No

7. PLOS authors have the option to publish the peer review history of their article (what does this mean?). If published, this will include your full peer review and any attached files.

Reviewer #1: Yes: Alan Campbell Kay

---

## [Editor Report · Acceptance letter]

5 Jun 2020

PONE-D-20-06821R1 

Epidemiology of hepatitis B and C virus infections among patients who booked for surgical procedures at Felegehiwot referral hospital, Northwest Ethiopia 

Dear Dr. Asemahagn:

I'm pleased to inform you that your manuscript has been deemed suitable for publication in PLOS ONE. Congratulations! Your manuscript is now with our production department. 

Kind regards, 

on behalf of

Mrs Isabelle Chemin 

Academic Editor

PLOS ONE